# The Influence of Macroclimatic Drivers on the Macrophyte Phylogenetic Diversity in South African Estuaries

## Dimitri Allastair Veldkornet

Department of Plant Sciences, Faculty of Natural and Agricultural Sciences, University of the Free State, P.O. Box 339, Bloemfontein 9300, South Africa; veldkornetda@ufs.ac.za

**Abstract:** The geographical distribution of plants is influenced by macroclimate and dispersal limitations, which have led to lineage isolation and subsequent diversification within and across various environmental gradients. Macroclimatic variables in coastal wetlands influence plant species and lineages across biogeographical boundaries. This study aimed to determine the influence of macroclimatic variables on species and phylogenetic richness in South African estuaries. Open-source chloroplast DNA barcoding sequences, species distribution and climatic data layers were used to determine the relationship between species richness, MPD, MNTD and each bioclimatic variable individually. Temperate species richness and phylogenetic diversity were positively correlated with temperature bioclimatic variables whereas subtropical and tropical species were associated with increases in precipitation. Phylogenetic niche conservatism is evident in malvids and rosids which are restricted to tropical and subtropical regions due to their physiological adaptations to tropical climates. Caryophylales was mostly associated with temperate regions. Poales and Alismatales showed wide distributions that is likely attributed to traits related to wind pollination and hydrochory, rapid, clonal, and high reproductive output, tolerance to stressful conditions, and intraspecific genetic diversity. The findings highlight the importance of considering macroclimate and phylogenetic factors in understanding the distribution and diversity of coastal wetland plants.

**Keywords:** coastal wetlands; climatic zones; phylogenetic niche conservatism; dispersal limitations; salt marsh; mangrove; coastal wetlands

## 1. Introduction

The primary purpose of DNA barcodes is for the identification of unknown samples [1,2]. However, the utilisation of DNA barcodes has extended to include species-level data to reconstruct evolutionary relations among phylogenetically disparate community members [3–7]. From this combined species community dataset and phylogenetic tree, phylogenetic diversity (PD) indices were developed as a measure of biodiversity that now incorporates the genetic differences between species to reflect how much evolutionary history exists within a community [8–10]. Indeed, this approach has been applied to various ecosystems to understand the process of community structuring [11–13], and other functional processes such as competition [14,15], environmental filtering [16], dispersal limitation [17], facilitation [18], alien invasion [7,19], predation [20], parasitism [21] and restoration [15].

The geographical distribution of plants is the result of the isolation of lineages due to historical factors, such as plate tectonics and dispersal limitations, that further resulted in their subsequent diversification across major environmental gradients [22,23]. Ecologists and evolutionary biologists have placed significant emphasis on unravelling the relative importance of these macroclimatic factors (temperature and precipitation) and geographic isolation in shaping the spatiotemporal assembly of plant communities [22,23]. Biogeographers have long been studying the latitudinal diversity gradient (LDG), where plant species richness is highest in the warm, moist tropics and decreases towards the

cold, dry poles [24–26]. The precipitation and temperature gradients between tropical and temperate regions are characterised by predictable decreases and increases in seasonal and daily temperature variations [27–31]. The capacity of plants to tolerate extreme ranges and fluctuations in rainfall and temperature are important drivers of plant species distributions along gradients and phylogenetic niche conservatism [31].

The phylogenetic niche conservatism hypothesis has been proposed to explain this distinct pattern [26,31–33]. Speciation creates closely related groups of related species that show similar macroenvironmental preferences within each group, and therefore, adaptive divergence between distinct macroenvironments occurs infrequently [31]. The close evolutionary relationship between species often leads to the inheritance of conservative similar traits from their common ancestors that is the result of environmental requirements, and tolerance to specific conditions. Tropical species, for example, are adapted to a narrow and stable range of abiotic conditions, which would explain why only a few species and lineages have managed to expand beyond the tropics [26,31,34,35].

Although not fully explored, on the global and regional scale the effect of evolutionary niche conservatism would be expected to be pronounced in coastal wetlands that are divided into different bioclimatic zones. It has been found that macroclimatic variables in coastal wetlands greatly influence the distribution of plant species across biogeographical boundaries [27,29,30,36]. Furthermore, biogeographic studies have primarily focused on the drivers of taxonomic diversity, with little to no attention given to the influence of macroclimatic variables on the phylogenetic and functional diversity of regional assemblages. Nevertheless, adopting this approach holds promising potential to reveal valuable insights into the eco-evolutionary processes that influence phylogenetic assemblages. In addition, there is a sparsity of information on whether climate change may result in the loss of phylogenetic diversity (PD). For example, it was found that for Californian plants, the phylogenetic diversity of the woody flora increased with increasing mean annual rainfall [30,36]. The authors also highlighted that there is a lack of understanding of the turnover of tropical plant lineages due to the restrictiveness of most studies focusing on areas with limited geographical extent, as well as the paucity of species distribution data or phylogenies [37]. The same caveats exist in coastal ecosystems where most studies are geared toward a regional setting (e.g., [38–41]), specific taxa (e.g., [42]) and whereas others do not include bioclimatic variables (e.g., [27,29,40]).

The application of biogeographic and phylogenetic diversity studies may also be useful to determine how climate change (temperature and precipitation) will influence the distribution and adaptation ability of coastal species. The prevailing prediction is that mangrove and coastal swamp forests will likely expand towards the south, resulting in a decline of salt marsh habitats [27,29,30,36]. The same disruption in habitat distribution is anticipated for many other regions, particularly where there is a tropical-temperate climatic gradient, such as in Australia, New Zealand, South America, western North America, southeastern Africa, the Middle East, and Asia [30]. For example, in the Southern African region climate change will vary across different climatic zones. The tropical and subtropical climatic zone along the eastern coastline of South Africa will experience higher temperatures and rainfall, while the cool temperate climatic zone along the western coastline will have lower rainfall. Such changes are likely to affect the essential components of estuarine functioning, including river inflow and sediment depositions which will ultimately impact the microclimatic conditions of various species [29,30,43].

This study aimed to determine the influence of macroclimatic variables on species and phylogenetic richness in South African estuaries. The South African coastal environment provides an ideal landscape to evaluate the phylogenetic niche conservatism hypothesis. This is because the country has all three of the global coastal wetland climatic zones: The tropics and subtropics that is warm and humid are often dominated by swamp and mangrove forest; the cold and wet temperate climatic zone is composed of salt marshes of the graminoid type, and the arid and semi-arid climatic zone where saltmarshes succulent plants are most abundant [43,44]. Each of these regions may potentially support unique

clades that diversified in response to dispersal limitations and trait adaptations with major climatic zones.

There are also good DNA barcoding sequences [40], species distribution data [45] and climatic data layers available that are open source. Therefore, this study also shows how the utilisation of a combination of freely available data are used to study various hypotheses related to coastal biogeography and the potential conservation application thereof.

## 2. Materials and Methods

### 2.1. Study Area, Data Acquisition, and Species and Biogeographical Classification

The study area encompasses the entire South African coastline (Figure 1). The estuaries have previously constituted three climatic zones namely Subtropical, Warm Temperate and Cool Temperate [46]. Recently, a new climatic zone has been designated Tropical because certain species in the Kosi and uMgobezeleni Estuaries in the northern region indicate tropical affinities [47]. This observation served as a strong encouragement to divide the Subtropical region into distinct parts, incorporating a tropical transition zone in the northeastern part of South Africa. Botanically, the region hosts unique species such as the mangroves, *Lumnitzera racemosa*, and *Ceriops tagal*. The climatic zones (biogeographical regions, per [47]) are now classified as tropical from Kosi to uMgobezeleni, the subtropical that stretches from St Lucia in KwaZulu-Natal to the Mbashe Estuary in the Eastern Cape, the warm-temperate from the Mendwana Estuary to the Ratel Estuary near Cape Agulhas, and the cool-temperate from the Uilkraals Estuary to the Orange Estuary on the Northern Cape coast. The Botanical Database of South African Estuaries contains records of the distribution of macrophyte species [45]. Using multivariate analysis, the authors were able to separate species according to different climatic zones (see [45] for a full description). The same database was used to determine the distribution of species in relation to bioclimatic variables in this study. In total 53 species were used to construct a community dataset (absence and presence) distributed across the four climatic zones.

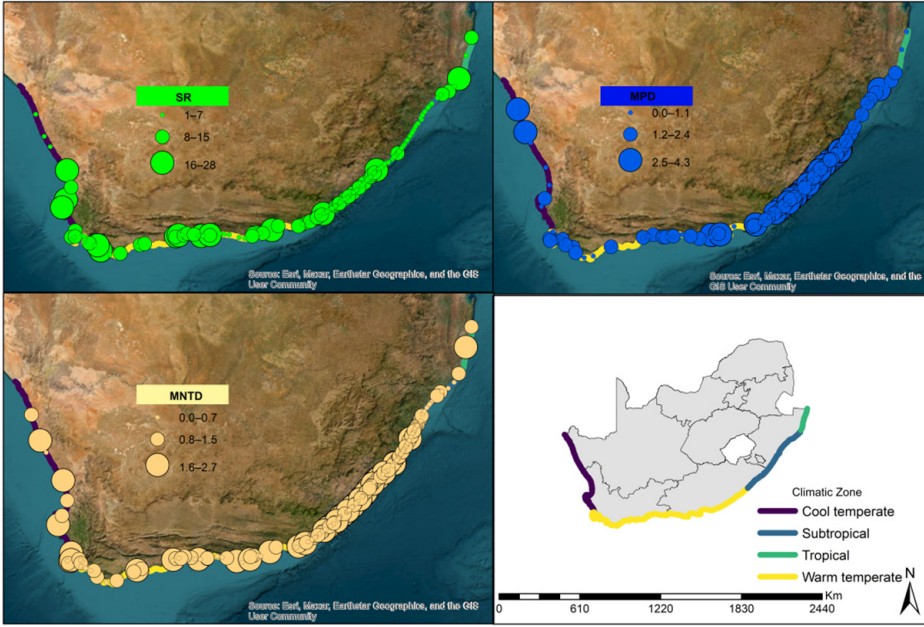

**Figure 1.** The distribution of species richness and phylogenetic diversity across different climatic zones in South Africa. Maps are based on the WGS 1984 Albers projected coordinate system. The frames show species richness (SR) and the two measures of phylogenetic diversity. The size of the circles indicates the relative values of each measurement. SR = Species richness, MPD = The average of the distances between all pairs of species in the community, MNTD = The average of the distances between each species and its nearest relative in the community.

## 2.2. Macroclimatic Analyses

To explore the bioclimatic variables that may influence the distribution of the species and phylogenetic clades the assumption was that the current species or community distribution is in equilibrium with its environment [48]. For this reason, climate information was obtained from the WorldClim database (version 1.4; https://www.worldclim.org/data/worldclim21.html, accessed on 20 February 2023). These climatic layers (19) are based on weather conditions recorded over 50 years from 1950 to 2000 on a grid of 30 s resolution. Veldkornet and Rajkaran [43] used bioclimatic variables in combination with sea surface temperature (SST) in species distribution modelling. They found that bioclimatic variables are better predictors compared to SST, perhaps because of the semi-aquatic nature of the plant species that are influenced by coastal climates. For this reason, SST was not included in this study.

Multicollinearity between predictor variables was assessed using the variance inflation factor (VIF) based on the square of the multiple correlation coefficient resulting from regressing a predictor variable against all other predictor variables. Multicollinearity was assessed using the vifcor function in the R package usdm (version 2.1-6) and variables with a threshold greater than 0.7 were excluded from further analysis [49]. This resulted in nine bioclimatic variables being selected: Mean Diurnal Range (°C), Isothermality (°C), Temperature Seasonality (°C), Minimum temperature of coldest month (°C), Temperature annual range (°C), Mean temperature of wettest quarter (°C), Annual precipitation (mm), Precipitation of wettest month (mm), Precipitation of wettest quarter (mm).

To determine the differences in bioclimatic variables between different climatic zones a Kruskal-Wallis multiple comparison with Bonferroni corrections was conducted, using the package agricolae (version 1.3-6) in R. To determine the relationship between species presence and each bioclimatic variable individually, linear regression models were fitted and *p*-values using Bonferroni correction in the R package ggpmisc (version 0.5.4-1). To determine the association of species and bioclimatic variables in relation to different biogeographical regions detrended correspondence analysis was conducted using the decorana, with Hellinger transformation standardisation, in vegan (version 2.6-4).

## 2.3. Phylogenetic Diversity Analyses

To determine the phylogenetic diversity of individual estuaries a previously published time-calibrated phylogeny of chloroplast DNA (*rbcLa* + *matK* sequences) was used to infer the evolutionary history of 47 South African estuarine macrophyte species [5]. Species names and their distribution in different climatic zones are in Appendix A (Table A1) and GenBank Accession numbers are presented in the Supplementary Materials (Table S1). The differences in the number of species between the community dataset and the phylogeny are due to the presence of one species in an estuary that would not allow for the determination of phylogenetic diversity in that estuary.

Two other measures of PD were calculated for each estuary. The first measure, Mean Pairwise Distance (MPD), represents the average phylogenetic distance (branch length) between all pairs of species within a given community. The second measure, Mean Nearest Taxon Distance (MNTD), represents the average distance between each species within a community and its closest relative. The calculations of MPD and MNTD were based on a distance matrix and a phylogeny object, using a combination of the *rbcLa* + *matK* datasets. To account for potential correlations of metrics with species richness, richness-independent standardized effect sizes were computed for each metric, resulting in $_{SES}$MPD and $_{SES}$MNTD. To assess whether individual estuaries exhibited significant phylogenetic clustering or overdispersion, the MPD and MNTD values were compared to null mean values. These null values were generated by randomising species across the tips of the phylogeny 999 times. This process allowed for the determination of whether the observed PD values were significantly different from random expectations. In the current study Phylogenetic diversity (PD) [8], the sum of the branch lengths of a phylogenetic tree connecting all species or taxa in a community was not determined as it was previously

shown to show significantly strong positive correlations with species richness (PD, r = 0.93; $p < 0.000$; [5]). The 'Picante' package (version 1.8.2) in the R programming language was used for this analysis [50]).

### 2.4. Phylogenetic Diversity Patterns in Relation to Bioclimatic Variables

To determine the relationship between species richness, MPD, MNTD and each bioclimatic variable individually, linear regression models were fitted, and *p*-values determined using Bonferroni correction. Only bioclimatic variables that were significantly correlated with MPD and MNTD were included in further analyses these were Mean Diurnal Range (°C), Isothermality (°C), Temperature annual range (°C), Minimum temperature of the coldest month (°C). Annual precipitation (mm), Precipitation of wettest month (mm) and Precipitation of wettest quarter (mm).

### 3. Results

Bioclimatic variables related to temperature and precipitation varied across climatic zones (Table 1).

**Table 1.** Mean (maximum and minimum) values of the bioclimatic variables in the different climatic zones.

| Bioclimatic Variable | Cool Temperate | Warm Temperate | Subtropical | Tropical |
| --- | --- | --- | --- | --- |
| Mean Diurnal Range (°C) | 9.91 (7.2; 13.4) | 7.86 (7.1; 10) | 10.4 (10.3; 10.5) | 8.8 (6.8; 11.8) |
| Isothermality (°C) | 54.48 (43; 64) | 55.40 (49; 60) | 53.23 (50; 56) | 56.5 (55; 58) |
| Temperature Seasonality (°C) | 28.44 (19.25; 41.03) | 22.96 (19.16; 31.62) | 22.59 (19.61; 28.12) | 24.98 (23.6; 26.36) |
| Minimum temperature of coldest month (°C) | 0.80 (0.72; 1.02) | 0.95 (0.63; 1.15) | 1.19 (1.12; 1.32) | 1.25 (1.2; 1.29) |
| Temperature annual range (°C) | 1.80 (1.37; 2.16) | 1.572 (1.37; 2.13) | 1.47 (1.28; 1.84) | 1.87(1.77; 1.89) |
| Mean temperature of the wettest quarter (°C) | 1.35 (1.2; 1.56) | 1.79 (1.36; 2.17) | 2.30 (2.15; 2.52) | 2.54 (2.52; 2.55) |
| Annual precipitation (mm) | 526 (93; 873) | 768 (424; 1087) | 1078 (966; 1150) | 924 (916; 932) |
| Precipitation of wettest month (mm) | 84 (17; 143) | 90 (46; 147) | 138 (119; 148) | 144 (139; 150) |
| Precipitation of wettest quarter (mm) | 237 (45; 399) | 241(126; 365) | 379 (352; 401) | 374 (374; 375) |

The multivariate analysis (DCA) of estuaries (sites) and bioclimatic variables in the different South African estuarine climatic zones are presented in Figure 2. The first two axes (DCA1 and DCA2) account for 63.05% of the total variation. Estuaries were separated in ordination space where tropical estuaries were separated from cool and warm temperate estuaries, subtropical estuaries were separated from cool temperate estuaries. Univariate analysis showed that all bioclimatic variables have significantly influenced the associations of species ($p < 0.05$). Temperate species were positively correlated with temperature bioclimatic variables (temperature diurnal range, r = 0.12; Isothermality, r = 0.12; minimum temperature of the coldest month, r = 0.1; temperature annual range, r = 0.55; temperature of the wettest quarter, r = 0.46; $p = 001$), whereas subtropical and tropical species were associated with increases in precipitation (annual precipitation, r = 0.46; precipitation of the wettest month, r = 0.48; precipitation of the wettest quarter, r = 0.52; precipitation of the warmest quarter r = 0.54; $p = 001$).

In total eight (8) clades representing 47 estuarine plant species are represented in the phylogenetic tree (Figure 3). Clades that were found in all climatic zones are Alismatales and Poales. No clades were restricted to the cool temperate climatic zone. Caryophyllales was mostly associated with the temperate regions, except for one species (*Salicornia polystachia*). The malvids clade (*Hibiscus tiliaceus*) was only associated with the tropical climatic zone. Polypodiales (*Acrostichum aureum*) was only associated with the tropical climatic zone. Except for *Avicennia marina* (warm temperate, subtropical, and tropical distribution) and *Barringtonia racemosa* (subtropical), most of the Asterid clade was found in the temperate regions. Charales was restricted to the temperate regions. All rosids were restricted to the tropical (*Lumnitzera racemosa*) and subtropical biogeographic (*Bruguiera gymnorrhiza* and *Rhizophora mucronate*, subtropical and tropical distribution).

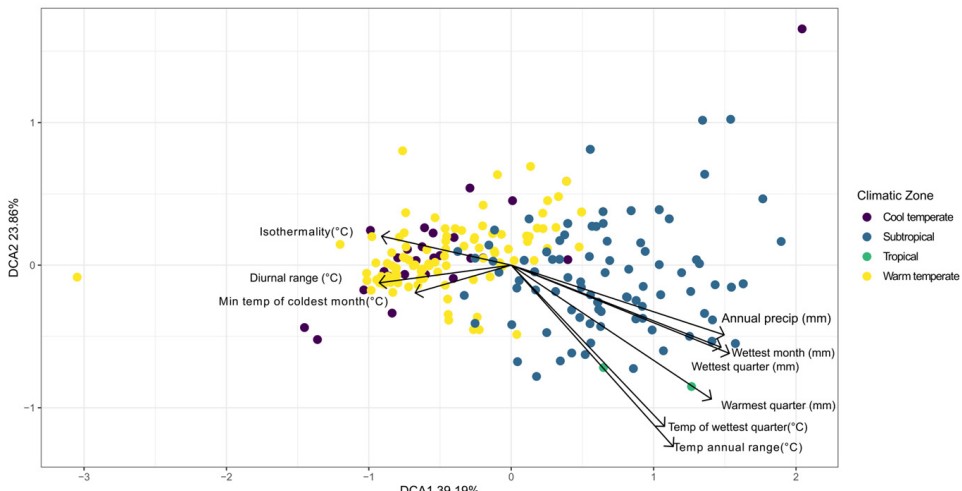

**Figure 2.** Detrended correspondence analysis (DCA) of estuarine plant estuaries and bioclimatic variables in relation to different climatic zones. Eigenvalues DCA1 = 0.4750, DCA2 = 0.2893. The sum of all the Eigenvalues = 1.2047.

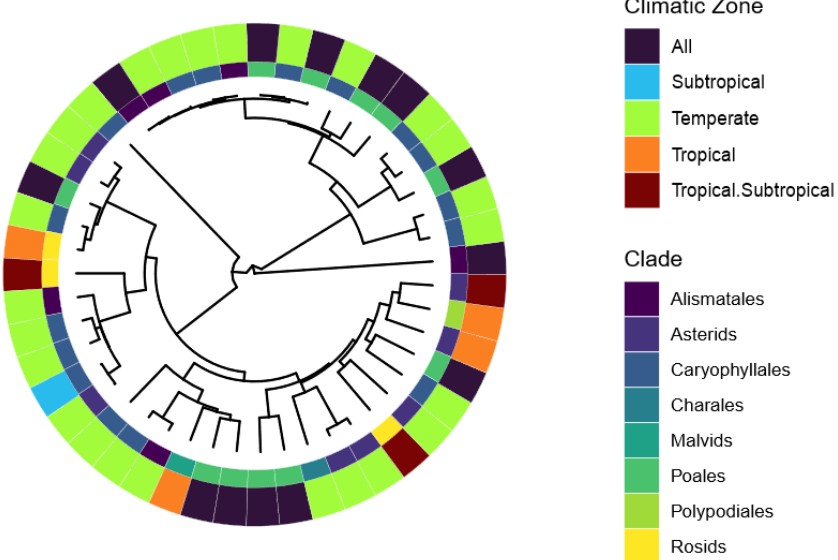

**Figure 3.** Phylogenetic tree of species and clades associated with the different climatic zones included in this study. The outer circle represents the climatic zones, and the inner circle represents the different clades. This tree was visualized with ggtree (version 3.2.1) using the function gheatmap in R.

The results of the regression analysis of phylogenetic diversity and bioclimatic variables are presented in Figure 4. There was a significantly negative correlation between MNTD and annual precipitation ($p = 0.0137$). MNTD significantly decreased with precipitation of the wettest month ($p = 0.030$) where the highest values of precipitation were associated with the tropical regions. MNTD also significantly decreased with precipitation of the wettest quarter ($p = 0.008$) where the highest values of precipitation were associated with the tropical regions. MPD significantly decreased with the mean diurnal range ($p = 0.010$) with the highest MPD found in the subtropical estuaries associated with the low diurnal range. In contrast, cool temperate estuaries were associated with higher diurnal ranges. Phylogenetic diversity significantly increased with isothermality (MPD, $p = 0.034$; MNTD, $p = 0.001$) with cool temperate estuaries associated with higher isothermality. Phylogenetic diversity (MPD) significantly increased with an increase in the minimum temperature of the coldest month ($p = 0.014$) where cool and warm temperate estuaries were associated with lower minimum temperatures compared to subtropical and tropical

estuaries. MPD significantly decreased with temperature annual range where the highest MPD was found for subtropical estuaries. Cool temperature estuaries were associated with higher temperature ranges.

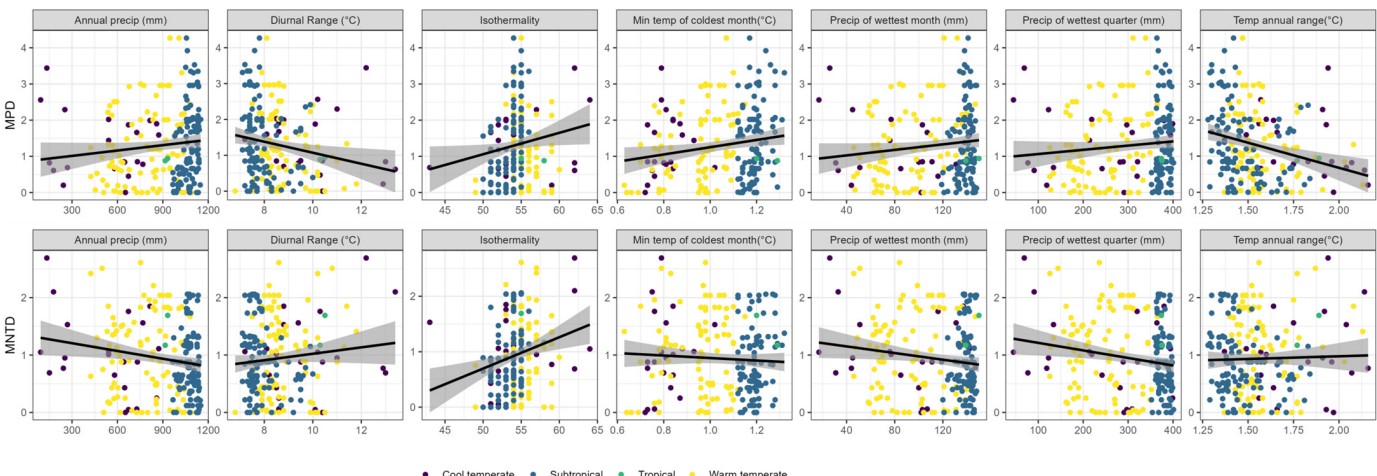

**Figure 4.** The relationship between phylogenetic diversity (MPD and MNTD) with bioclimatic variables. Mean Diurnal Range (°C), Isothermality (°C), Temperature Seasonality (°C), Minimum temperature of coldest month (°C), Temperature annual range (°C), Mean temperature of wettest quarter (°C), Annual precipitation (mm), Precipitation of wettest month (mm), Precipitation of wettest quarter (mm) and Precipitation of warmest quarter (mm).

## 4. Discussion

The global distribution of coastal wetlands is governed by the macroclimate that creates large-scale climatic zones where they can generally be divided into three bioclimatic zones: the tropics, cool temperate and dry climatic zones [20,29,51–54]. This study provides empirical evidence for the biogeographical separation of South African tidal coastal wetlands into at least three climatic regions and the climatic variables that are responsible for these patterns. This study explores the climatic association of species in relation to bioclimatic variables without describing the species associated with each of these. A description of the species in the different climatic regions in South Africa can be found in [45].

The separation of species and estuaries in ordination space into tropical, subtropical, cool warm temperate, can be directly attributed to variations in temperature and precipitation across the South African coastline. On the biogeographic scale (macroecology) temperature and rainfall have been deemed the most important factors affecting estuarine species distribution and evolutionary change [52,55,56]. Thresholds of temperature and precipitation are known to dictate the physiological response of plants (see [57,58] for a complete review). Plants can experience stress and reduced metabolic activity due to extreme temperatures like heatwaves or frost events [58–60]. Water uptake is also critical for photosynthesis and nutrient transport, and rainfall plays a crucial role in this process [56]. Conversely, insufficient rainfall can lead to drought stress [61,62] and frost, negatively impacting stomatal conductance, nutrient uptake, and overall plant metabolism [62]. Therefore, the positive association of temperate species with high variations in the mean diurnal range and isothermality can be attributed to adaptation to temperature variability. Temperature diurnal range refers to the difference between the maximum and minimum temperatures that occur within 24 h and isothermality refers to the degree of temperature variation throughout the year. Plants that are adapted to temperate climates tend to do well in areas with significant temperature differences between day and night. These plants have developed various physiological and biochemical mechanisms to survive and thrive in changing temperatures and can, therefore, withstand both high daytime temperatures and lower nighttime temperatures [63]. In contrast, tropical and subtropical species richness were negatively influenced by temperature diurnal range. In tropical regions, the tempera-

ture diurnal range is often relatively small compared to temperate regions. Forests (such as mangrove and swamp forests) are also buffered from temperature extremes, and this is most prominent in summer months when understorey plants are most likely to experience extreme heat and drought stress [64,65]. Furthermore, a wide diurnal temperature range influences species diversity because some clades have evolved in tropical climates and cannot disperse into cold regions owing to their niche conservatism [63]. Therefore, the inability of tropical species in this study to tolerate temperature fluctuations throughout the day may be responsible for their lack of distribution into temperate climates.

The association between temperate species and the minimum temperature of the coldest month may reflect their ability to tolerate frost conditions. Unlike tropical plants, most temperate plant species can tolerate frost conditions [63]. Frost occurs when temperatures drop below freezing point, causing ice crystals to form in plant tissues and on surfaces, thereby becoming a limiting factor for plant survival and distribution [56]. At evolutionary time scales, frost and freezing temperatures have posed significant barriers to the establishment and survival of tropical woody evergreen plants [26]. Many tropical clades face obstacles in colonising areas where frost occurs [26]. Frost also poses a major barrier to dispersal and adaptation of most terrestrial and marine organisms, and areas experiencing frost, resulting in high phylogenetic turnover among continents and across the tropical-temperate divide, which manifests in tropical niche conservatism [66].

A significant relationship was also found between the presence of cool temperate species and the temperature of the wettest quarter. The temperature of the wettest quarter is an important climatic factor that can significantly influence species distribution, particularly in temperate regions with distinct wet and dry seasons. In South Africa, cool temperate estuaries are characterised by high salinity and low turbidity because of low rainfall and runoff, high seawater input and evaporative loss [5,67]. Higher temperatures during the wettest quarter can also lead to increased evaporation of soil moisture between rain events. As water evaporates, it leaves behind dissolved salts in the soil, resulting in higher soil salinity. In cool temperate estuaries such as the Orange, Olifants, Berg and Verlorenvlei estuaries the average annual precipitation is approximately 50 mm per year, with an average potential evaporation of over 3000 mm per year [45]. In the newly described South African estuary type, arid predominantly closed estuaries, in the cool temperate regions salt tolerant, succulent *Salicornia* spp. can live in open water salinities reaching greater than 200 ppt [47,68].

Subtropical and tropical species were associated with increases in precipitation (annual precipitation, precipitation of the wettest month, precipitation of the wettest quarter, precipitation of the warmest quarter). Precipitation is the primary axis influencing the distribution of tropical plants [66]. Studies have also found that higher species diversity is found in low-altitude areas such as the tropics (e.g., [26,36,59,63,66]) where species diversity of trees significantly increased with precipitation and significantly decreased with climate variability. It can, therefore, be assumed that the presence of woody species in estuarine habitats such as mangroves and swamp forests is in relation to water availability and temperature stability.

The phylogenetic niche conservatism hypothesis states the observed patterns of phylogenetic relatedness among species and their ecological niches tend to retain ancestral ecological traits and have similar niche requirements [69–71]. According to this hypothesis, the phylogenetic niche conservatism (PNC) of ancestral traits adapted to tropical conditions has significantly influenced the phylogenetic structure and composition of regional species pools. The ability of clades to persist in different regions is largely determined by the minimum temperatures experienced in those regions [23,31,36,59,71]. In the current study, it was found that three clades were restricted to the tropical and subtropical (malvids, Polypodiales, rosids), two clades (Caryophylales and asterids) were mostly associated with the temperate regions, and two clades (Alismatales and Poales) had species distributed across all four climatic zones.

The distribution of malvids and rosids in the tropical and subtropical climatic zones can be explained by the warm and humid climates, which provide favourable conditions for the growth and proliferation of diverse plant species. In malvids, *Hibiscus tiliaceus* (Malvaceae), occurs only in subtropical estuaries. This could be due to phylogenetic niche conservatism in the tribe with populations not being able to survive the cooler temperatures at higher altitudes. This species favours environments with uniform rainfall (900–2500 mm) patterns and can tolerate a minimum temperature of 14 °C, characteristic of tropical temperatures [43]. It has been suggested that Malvaceae evolved in the tropical ancestors but later diversified at a rate comparable with many tropical lineages in the family [72]. These results are consistent findings that species of Malvaceae are tropical and the family thus conforms to a latitudinal gradient in species diversity. In a climate change modelling approach, low habitat suitability and potential distribution were found for *H. tiliaceus* in warm temperate estuaries because future minimum temperatures (2050) will not increase above the minimum temperature tolerance of the species [43].

Polypodiales is composed of all major polypod ferns, and like *H. tiliaceus*, *Acrostichum aureum* is also restricted to the tropical climatic zone. It was found that the diversity of fern species increased with decreasing latitude and with increasing temperature and precipitation which is congruent with the tropical niche conservatism hypothesis, indicating physiological adaptations to tropical climates [73,74].

The rosids clade was composed of mangrove species that were restricted to the tropical and subtropical climatic zones. The restriction of rosids to the tropics supports phylogenetic niche conservatism which predicts that few lineages can colonise and radiate in colder and/or drier regions. This will produce clusters of phylogenetically closely related species. It was also found that Rosaceae species richness increased significantly with mean annual temperature, mean temperature of the coldest quarter and mean annual precipitation significantly decreased with temperature annual range [59]. Mangroves in the rosids group are particularly sensitive to low temperatures where the species are limited to the 16 °C air temperature isotherm of the coldest month [75]. In contrast to the mangroves in the rosids clade, *Avicennia marina* (asterids) can tolerate colder temperatures due to its ability to recover from extreme winter air temperatures and responses to chilling and freezing conditions [75].

The asterids clade had the highest species richness, particularly, Asteraceae in temperate regions. Most of these species were associated with salt marshes. Asterids generally have a wide ecological niche, allowing them to tolerate a broad range of environmental conditions. This adaptability enables the clade to thrive in different climatic zones, including both tropical and subtropical regions. However, the greatest diversity of asterids occurs in the arid and semi-arid regions compared to subtropical and lower temperate latitudes [76]. In estuaries, asterids are also found in different estuarine habits, for example, *Cotula coronopifolia* is found in the lower intertidal zone, whereas *Samolus porosus* is found in the upper intertidal zone, suggesting ecological niche specialisation [45,77]. The drivers of such ecological niche differentiation along a tidal elevation gradient should be explored from a phylogenetic perspective.

The species richness and restrictiveness of Caryophylales to the temperate zone can be explained by the adaptations of this clade to salt marsh conditions [77]. In this study, the clade was represented by 16 species in five families, with the genus *Salicornia* being the most diverse. The centre of diversity of the genus is in southern Africa where 12 species are found in salt marshes with high soil salinity due to seasonal lack of precipitation, frost, and frequent and often prolonged flooding [70]. The genus also shows ecological niche differentiation of taxa along a salt marsh elevation [78]. Approximately 21.4% of halophytic angiosperm species can be found in nine families of Caryophyllales. Among these families, Amaranthaceae has the highest number of halophytes [79]. The common ancestor of the subfamilies Salicornioideae, Suaedoideae, Camphorosmoideae, and Salsoloideae, which existed 61–35 million years ago, also had salt tolerance [80]. It can, therefore, be concluded that the phylogenetic niche conservatism in halophytes is associated with low annual

rainfall and high evaporation rates that favour the retention of similar adaptive traits related to saline environments among closely related species.

This study found two clades, Poales and Alismatales, that did not exhibit phylogenetic niche conservatism and were present in all four climatic zones. This is likely because these clades contain species with varying dispersal abilities. Some species have efficient mechanisms for long-distance dispersal, allowing them to establish new populations in different environments and adapt to new habitats [81]. The wide distribution of Poales can be attributed to several traits such as wind pollination and seed dispersal, rapid, clonal, and high reproductive output, and tolerance to stressful and disturbed environments [81,82]. It was also found that coastal species in Poales have high haplotype diversity compared to their terrestrial conspecifics suggesting a genetic adaptation for along a climatic diverse coastline [81–83]. Similar to Poales, Alismatales exhibit long-distance dispersal traits (e.g., [18], for seagrasses). In Southern Africa, the seagrass *Zostera capensis* can establish through a persistent rhizome structure and vegetative re-growth [84]. However, in contrast to Poales which exhibit high haplotype diversity, the dispersal of clonal parts via ocean current may also be responsible for the low genetic diversity that is observed in this species [85,86]. Also, within Alismatales, the species *Triglochin bulbosa* was only found in the temperate climatic zone, whereas *T. striata* was found in all climatic zones. Von Mering and Kadereit (2015) [87] suggested that the dispersal by sea currents or birds seems a likely explanation for the wide distribution of *T. striata*. All these traits can reduce the expression of phylogenetic niche conservatism of these species.

It has been observed that regions with higher levels of annual precipitation also have higher levels of phylogenetic diversity (MPD). In the subtropical and tropical climatic zones, where water is essential for plant growth and survival, increased precipitation often results in greater water availability. This can support a greater number of plant species, leading to speciation and diversification, and ultimately resulting in higher MPD values. Additionally, areas with higher annual precipitation may experience more stable environmental conditions due to water availability [64,66].

A significant negative correlation between the temperature range, mean diurnal temperature range, isothermality and MPD was observed suggesting that regions such as the cool temperate biogeographic with higher temperature fluctuations tend to have lower PD values. In contrast, these arid areas also are associated with higher MPD values. These harsh environmental conditions can result in strong selective pressures on plant species, leading to the evolution and persistence of specialized and unique lineages. The ability of certain species to withstand extreme aridity and adapt to a wide range of arid habitats may contribute to higher MPD values [63,78–80].

MNTD, which measures the average distance between each species and its closest relative within the community showed a negative correlation with annual precipitation and more precipitation during the wettest month and wettest quarter. This suggests that the phylogenetic relatedness of co-occurring species in the community tends to decrease. These habitats were phylogenetically overdispersal possibly due to the presence of different clades (Polypodiales, asterids, rosids, malvids, Caryophyllales). Due to the benign conditions in the tropics resulting in higher species richness biotic interactions, such as competitive exclusion, are likely to play a dominant role in determining species coexistence [15]. Similar results were also found by [4], where along streams and in swamps, species assemblages were no different from those expected by chance, which suggests that phylogenetic relatedness may not play a role in determining the structure of communities in these habitats.

## 5. Conclusions

This provides valuable insights into the biogeographic distribution and climatic influences on coastal wetlands in South Africa. Empirical evidence is given for the influence of climatic variables on the biogeographic botanical separation estuaries and the patterns of species distribution in relation to phylogenetic relatedness. Moreover, temperature and rain-

fall are crucial factors influencing estuarine species distribution and evolutionary change at the macroecological scale. The temperature diurnal range and isothermality influence the distribution of temperate species, with higher values associated with temperate regions. In contrast, tropical and subtropical species are negatively influenced by temperature diurnal range and show limited niche conservatism. Phylogenetic niche conservatism is evident in certain clades, such as malvids and rosids, which are restricted to tropical and subtropical regions due to their physiological adaptations to tropical climates, or rather their inability to tolerate large fluctuations in temperatures. However, Poales and Alismatales showed a wide distribution, likely due to species with varying dispersal abilities. The findings highlight the importance of considering macroclimate and phylogenetic factors in understanding the distribution and diversity of coastal wetland plant species.

**Supplementary Materials:** The following supporting information can be downloaded at: https://www.mdpi.com/article/10.3390/d15090986/s1, Table S1: Selection of species for phylogenetic diversity analysis.

**Funding:** The original research that generated the DNA barcodes [5] was funded by the National Research Foundation through the South African Network for Coastal and Oceanic Research (SANCOR, Grant Number: SANG160510164566).

**Institutional Review Board Statement:** Not applicable.

**Data Availability Statement:** The DNA barcode accession numbers for each species are presented in the Supplementary Materials. The Estuarine Botanical Database, which contains macrophyte species distribution in SA estuaries is accessible at http://hdl.handle.net/20.500.12143/6707 (accessed on 20 February 2023. Climate variables are available in the WorldClim database (version 1.4; https://www.worldclim.org/data/worldclim21.html, accessed on 20 February 2023). R codes are available on request.

**Conflicts of Interest:** The author declares no conflict of interest. The funders had no role in the design of the study; in the collection, analyses, or interpretation of data; in the writing of the manuscript; or in the decision to publish the results.

## Appendix A

**Table A1.** Species names and their distribution in different climatic zones.

| Species | Family | Clade | Climatic Zone | Habitat |
|---|---|---|---|---|
| *Acrostichum aureum* | Polypodiaceae | Polypodiales | Tropical | Mangrove |
| *Avicennia marina* | Acanthaceae | Asterids | Tropical.Subtropical | Mangrove |
| *Barringtonia racemosa* | Lecythidaceae | Asterids | Tropical | Swamp Forest |
| *Bassia diffusa* | Amaranthaceae | Caryophyllales | Temperate | Supratidal Salt Marsh |
| *Bolboschoenus maritimus* | Cyperaceae | Poales | All | Reeds and Sedges |
| *Bruguiera gymnorrhiza* | Rhizophoraceae | Rosids | Tropical.Subtropical | Mangrove |
| *Centella asiatica* | Asteraceae | Asterids | Temperate | Intertidal Salt Marsh |
| *Chara vulgaris* | Charophyceae | Charales | Temperate | Submerged Macrophytes |
| *Nidorella ivifolia* | Asteraceae | Asterids | Temperate | Supratidal Salt Marsh |
| *Cotula coronopifolia* | Asteraceae | Asterids | Temperate | Intertidal Salt Marsh |
| *Cynodon dactylon* | Poaceae | Poales | All | Supratidal Salt Marsh |
| *Cyperus laevigatus* | Cyperaceae | Poales | All | Reeds and Sedges |
| *Disphyma crassifolium* | Aizoaceae | Caryophyllales | Temperate | Supratidal Salt Marsh |
| *Frankenia pulverulenta* | Frankeniaceae | Caryophyllales | Temperate | Supratidal Salt Marsh |
| *Halophila ovalis* | Hydrocharitaceae | Alismatales | Temperate | Submerged Macrophytes |
| *Hibiscus tiliaceus* | Malvaceae | Malvids | Tropical | Swamp Forest |
| *Isolepis cernua* | Cyperaceae | Poales | All | Reeds and Sedges |
| *Juncus acutus* | Juncaceae | Poales | All | Reeds and Sedges |
| *Limonium scabrum* | Plumbaginaceae | Caryophyllales | Temperate | Supratidal Salt Marsh |
| *Lumnitzera racemosa* | Combretaceae | Rosids | Tropical | Mangrove |
| *Phragmites australis* | Poaceae | Poales | All | Reeds and Sedges |
| *Plantago carnosa* | Plantaginaceae | Asterids | Temperate | Intertidal Salt Marsh |
| *Poecilolepis ficoidea* | Asteraceae | Asterids | Temperate | Supratidal Salt Marsh |

**Table A1.** *Cont.*

| Species | Family | Clade | Climatic Zone | Habitat |
|---|---|---|---|---|
| *Rhizophora mucronata* | Rhizophoraceae | Rosids | Tropical.Subtropical | Mangrove |
| *Ruppia cirrhosa* | Ruppiaceae | Alismatales | Temperate | Submerged Macrophytes |
| *Salicornia meyeriana* | Amaranthaceae | Caryophyllales | Temperate | Intertidal Salt Marsh |
| *Salicornia uniflora* | Amaranthaceae | Caryophyllales | Temperate | Intertidal Salt Marsh |
| *Salicornia pachystachya* | Amaranthaceae | Caryophyllales | Subtropical | Intertidal Salt Marsh |
| *Samolus porosus* | Theophrastaceae | Asterids | Temperate | Intertidal Salt Marsh |
| *Salicornia capensis* | Amaranthaceae | Caryophyllales | Temperate | Supratidal Salt Marsh |
| *Salicornia decumbens* | Amaranthaceae | Caryophyllales | Temperate | Intertidal Salt Marsh |
| *Salicornia natalensis* | Amaranthaceae | Caryophyllales | Temperate | Intertidal Salt Marsh |
| *Salicornia pillansii* | Amaranthaceae | Caryophyllales | Temperate | Supratidal Salt Marsh |
| *Salicornia tegetaria* | Amaranthaceae | Caryophyllales | Temperate | Intertidal Salt Marsh |
| *Schoenoplectus scirpoides* | Cyperaceae | Poales | All | Reeds and Sedges |
| *Schoenoplectus triqueter* | Cyperaceae | Poales | All | Reeds and Sedges |
| *Spartina maritima* | Poaceae | Poales | All | Intertidal Salt Marsh |
| *Spergularia media* | Caryophyllaceae | Caryophyllales | Temperate | Supratidal Salt Marsh |
| *Spergularia rubra* | Caryophyllaceae | Caryophyllales | Temperate | Supratidal Salt Marsh |
| *Sporobolus virginicus* | Poaceae | Poales | All | Supratidal Salt Marsh |
| *Stenotaphrum secundatum* | Poaceae | Poales | All | Supratidal Salt Marsh |
| *Stuckenia pectinata* | Potamogetonaceae | Alismatales | Temperate | Submerged Macrophytes |
| *Suaeda inflata* | Amaranthaceae | Caryophyllales | Temperate | Supratidal Salt Marsh |
| *Suaeda fruticosa* | Amaranthaceae | Caryophyllales | Temperate | Supratidal Salt Marsh |
| *Triglochin bulbosa* | Juncaginaceae | Alismatales | Temperate | Intertidal Salt Marsh |
| *Triglochin striata* | Juncaginaceae | Alismatales | All | Intertidal Salt Marsh |
| *Zostera capensis* | Zosteraceae | Alismatales | All | Submerged Macrophytes |

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
