# Peer review of "The Influence of Macroclimatic Drivers on the Macrophyte Phylogenetic Diversity in South African Estuaries"

_diversity, doi:10.3390/d15090986_

Round 1

Reviewer 1 Report

The manuscript submitted for review concerns the issue of the influence of macroclimatic drivers on the macrophyte phylogenetic diversity in estuaries. Due to the climate changes observed all over the world, such subjects are needed. Overall, the manuscript is well laid out and conforms to publishing requirements. It also contains relevant information that expands scientific knowledge. Before publishing, however, it requires some corrections. Below are detailed notes: 1. Title - please specify what region of the world you are talking about 2. The Introduction should contain information on the influence of macroclimatic drivers, changes taking place in the world, etc. 3. It should be shown (e.g. in the form of a table) what changes in the microclimate were observed - with specific numerical values. 4. Fig. 4 is illegible

Author Response

Dear Editors

BelowI have attended to comments and suggestions by Reviewer 1. The reviewer is thanked for the useful suggestions. 

The manuscript submitted for review concerns the issue of the influence of macroclimatic drivers on the macrophyte phylogenetic diversity in estuaries. Due to the climate changes observed all over the world, such subjects are needed. Overall, the manuscript is well laid out and conforms to publishing requirements. It also contains relevant information that expands scientific knowledge. Before publishing, however, it requires some corrections. Below are detailed notes: 

  1. Title - please specify what region of the world you are talking about 

The title has been changed to “The Influence of Macroclimatic Drivers on the Macrophyte Phylogenetic Diversity in South African Estuaries”

  1. The Introduction should contain information on the influence of macroclimatic drivers, changes taking place in the world, etc. 3.

A section has been added to the introduction:

The application of biogeographic and phylogenetic diversity studies may also be useful in determining how climate change (temperature and precipitation) will influence the distribution and adaptation ability of coastal species.  The prevailing prediction is that mangrove and coastal swamp forests will likely expand towards the south, resulting in a decline of salt marsh habitats [30,29,27,36]. The same disruption in habitat distribution is anticipated for many other regions, particularly where there is a tropical-temperate climatic gradient, such as in Australia, New Zealand, South America, western North America, southeastern Africa, the Middle East, and Asia [30]. For example, in the Southern African region climate change will vary across different climatic zones. The tropical and subtropical climatic zone along the eastern coastline of South Africa will experience higher temperatures and rainfall, while the cool temperate climatic zone along the western coastline will have lower rainfall. Such changes are likely to affect the essential components of estuarine functioning, including river inflow and sediment depositions which will ultimately impact the microclimatic conditions of various species [29,30,43]. It has also been suggested that there is a need to understand how variation in climate will influence the distribution of species which is necessary to accurately predict how climate change will impact natural communities and ecosystems [30,36].

It should be shown (e.g. in the form of a table) what changes in the microclimate were observed - with specific numerical values. 

A table has been added to the main manuscript (Table 1) that describes the mean (maximum and minimum) values of the bioclimatic variables in the different climatic zones.

  1. Fig. 4 is illegible

This image is too large with the current layout. It is requested that the image should be adjusted during final processing.  

Reviewer 2 Report

This is an all-around good study, showing great breadth of skills on the part of the author. Well done!

Abstract: “Poales and Alismatales showed a wide distribution, likely due to 20 species with varying dispersal abilities.” I would have understood this if you said “due to good dispersal abilities”. As is, it needs some rephrasing of some kind or another. Same changes in relevant place in the Discussion.

Intro: To me this is a paper about estuarine macrophytes, not about barcoding. I would not start with a barcoding paragraph.

Fig 1 caption: Rephrase “the magnitude of each measurement”. Maybe just name it in each case.

Climatic zones in Fig 1 and Fig 2 should use same colour code. I would not call these bioregions anywhere in the paper.

In contrast, the two circles in Fig 3 should use different colour schemes.

Discussion: relevant to how various traits map on the phylogeny of plants, and how much phylogenetic conservatism different clades exhibit, see

https://www.researchgate.net/publication/372630580_Plants_used_by_humans_have_characteristics_that_exacerbate_invasions_worldwide

https://www.researchgate.net/publication/334392442_Soil_niche_of_rain_forest_plant_lineages_Implications_for_dominance_on_a_global_scale

and maybe also

https://www.researchgate.net/publication/366967345_What_determines_plant_species_diversity_along_the_Modern_Silk_Road_in_the_east

Appendix A: I would move family and clade ahead of bioregion (name to be changed) and habitat.

Author Response

Dear Editors 

Below I have attended to the comments and suggestions from Reviewer 2. The reviewer is thanked for the useful suggestions. 

This is an all-around good study, showing great breadth of skills on the part of the author. Well done!

Abstract: “Poales and Alismatales showed a wide distribution, likely due to 20 species with varying dispersal abilities.” I would have understood this if you said “due to good dispersal abilities”. As is, it needs some rephrasing of some kind or another. Same changes in relevant place in the Discussion.

The abstract has been rewritten as:

The geographical distribution of plants is influenced by macroclimate and dispersal limitations, which have led to lineage isolation and subsequent diversification across various environmental gradients. Macroclimatic variables in coastal wetlands influence plant species and lineages across biogeographical boundaries. This study aimed to determine the influence of macroclimatic variables on species and phylogenetic richness in South African estuaries. Open-source chloroplast DNA barcoding sequences, species distribution data and climatic data layers were used to determine the relationship between species richness, MPD, MNTD and each bioclimatic variable individually, linear regression models were fitted Bonferroni correction. Temperate species and phylogenetic diversity were positively correlated with temperature bioclimatic variables whereas subtropical and tropical species were associated with increases in precipitation. Phylogenetic niche conservatism is evident in malvids, rosids, and asterids, which are restricted to tropical and subtropical regions due to their physiological adaptations to tropical climates. Caryophylales was mostly associated with temperate regions. Poales and Alismatales showed wide distributions that likely attributed to traits related to wind pollination and hydrochory, rapid, clonal, and high reproductive output, and tolerance to stressful, and intraspecific genetic diversity. The findings highlight the importance of considering macroclimate and phylogenetic factors in understanding the distribution and diversity of coastal wetland plants.

 Intro: To me this is a paper about estuarine macrophytes, not about barcoding. I would not start with a barcoding paragraph.

I have included the section on barcoding because the result from this paper is based on barcoding data. The inclusion of the paragraph is to align barcoding with phylogenetic diversity and how it relates to biodiversity which further aligns with the theme of the special issue. It is therefore my opinion that the section should not be removed.

 Fig 1 caption: Rephrase “the magnitude of each measurement”. Maybe just name it in each case.

The caption has been rewritten as:

Figure 1. The distribution of species richness and phylogenetic diversity across different biogeographic regions in South Africa. Maps are based on the WGS 1984 Albers projected coordinate system.  The frames show species richness (SR) and the two measures of phylogenetic diversity. The size of the circles indicates the relative values of each measurement. SR = Species richness, MPD = The average of the distances between all pairs of species in the community, MNTD = The average of the distances between each species and its nearest relative in the community.

 Climatic zones in Fig 1 and Fig 2 should use the same colour code.

The colours have been changed in Fig 1 to match those of all the other figures.

I would not call these bioregions anywhere in the paper.

 Bioregions and biogeographical zones have been changed to climatic zones.

In contrast, the two circles in Fig 3 should use different colour schemes.

The colour schemes have been changed in Figure 3.

Discussion: relevant to how various traits map on the phylogeny of plants, and how much phylogenetic conservatism different clades exhibit, see

 https://www.researchgate.net/publication/372630580_Plants_used_by_humans_have_characteristics_that_exacerbate_invasions_worldwide 

https://www.researchgate.net/publication/334392442_Soil_niche_of_rain_forest_plant_lineages_Implications_for_dominance_on_a_global_scale 

and maybe also

https://www.researchgate.net/publication/366967345_What_determines_plant_species_diversity_along_the_Modern_Silk_Road_in_the_east

 This section on traits within the phylogeny of plants has been edited. It now reads:

This study found that two clades, Poales and Alismatales, did not exhibit phylogenetic niche conservatism and were present in all four climatic zones. This is likely because these clades contain species with varying dispersal abilities. Some species have efficient mechanisms for long-distance dispersal, allowing them to establish new populations in different environments and adapt to new habitats [81]. The wide distribution of Poales can be attributed to several traits such as wind pollution and seed dispersal, rapid, clonal, and high reproductive output, and tolerance to stressful and disturbed environments [81,82]. It was also found that coastal species in Poales have high haplotypes diversity compared to their terrestrial conspecifics suggesting a genetic adaptation for along a climatic diverse coastline [81,82,83]. Similar to Poales, Alismatales exhibit long-distance dispersal traits [e.g., 18, for seagrasses]. In Southern Africa, the seagrass Zostera capensis can establish through a persistent rhizome structure and vegetative re-growth [84]. In contrast to Poales which exhibit high haplotype diversity, the dispersal of clonal parts via ocean current may also be responsible for the low genetic diversity that is observed in this species [85, 86]. Also, within Alismatales, the species Triglochin bulbosa was only found in the temperate climatic zone, whereas T. striata was found in all climatic zones. Von Mering and Kadereit (2015) [87] suggested that the dispersal by sea currents or birds seems a likely explanation for the wide distribution of T. striata. All these traits can reduce the expression of phylogenetic niche conservatism of these species.

Appendix A: I would move family and clade ahead of bioregion (name to be changed) and habitat.

So, moved.

Round 2

Reviewer 1 Report

The Author has significantly improved his manuscript. It can be published in its present form.

Author Response

The reviewers are thanked for the comments on the manuscript.

Reviewer 2 Report

None, all good.

Author Response

The reviewer is thanked for the comments on the manuscript.